developing countries; assessment tools; psychometric evaluations; primary care; cross-cultural

**Corresponding author:**
Melissa Ann Stockton;
Email: mastockt@email.unc.edu

# Validation of a brief screener for broad-spectrum mental and substance-use disorders in South Africa

Melissa Ann Stockton[1] [iD], Ernesha Webb Mazinyo[2,14], Lungelwa Mlanjeni[2], Kwanda Nogemane[3], Nondumiso Ngcelwane[3], Annika C. Sweetland[4,5], Cale Neil Basaraba[6,7], Charl Bezuidenhout[8], Griffin Sansbury[9] [iD], Kathryn L. Lovero[10] [iD], David Olivier[11], Christoffel Grobler[12], Melanie M. Wall[4,5], Andrew Medina-Marino[11,13], Phumza Nobatyi[3] and Milton L. Wainberg[4,5]

[1]Department of Epidemiology, Gillings School of Global Public Health, University of North Carolina at Chapel Hill, Chapel Hill, NC, USA; [2]Research Unit, Foundation for Professional Development, Buffalo City Metro, Eastern Cape Province, South Africa; [3]Buffalo City Metro Health District, Eastern Cape Provincial Department of Health, Bisho, South Africa; [4]Department of Psychiatry, Columbia University Vagelos College of Physicians and Surgeons, New York, USA; [5]New York State Psychiatric Institute, New York, USA; [6]Department of Population and Family Health, Columbia University Mailman School of Public Health, New York, NY, USA; [7]Department of Biostatistics, Columbia University Mailman School of Public Health, New York, NY, USA; [8]Department of Global Health, Boston University School of Public Health, Boston, MA, USA; [9]University of North Carolina-Project, Malawi, Lilongwe, Malawi; [10]Department of Sociomedical Sciences, Columbia University Mailman School of Public Health, New York, USA; [11]The Desmond Tutu HIV Centre, University of Cape Town, Cape Town, South Africa; [12]Faculty of Medicine, School of Health Systems and Public Health, University of Pretoria, Pretoria, South Africa; [13]Perelman School of Medicine, University of Pennsylvania, Philadelphia, PA, USA and [14]University of California Global Health Institute, University of California, San Francisco, USA

## Abstract

In low-resource settings, valid mental health screening tools for non-specialists can be used to identify patients with psychiatric disorders in need of critical mental health care. The Mental Wellness Tool-13 (mwTool-13) is a 13-item screener for identifying adults at risk for common mental disorders (CMDs) alcohol-use disorders (AUDs), substance-use disorders (SUD), severe mental disorders (SMDs), and suicide risk (SR). The mwTool-13 is administered in two steps, specifically, only those who endorse any of the initial three questions receive the remaining ten questions. We evaluated the performance of mwTool-13 in South Africa against a diagnostic gold standard. We recruited a targeted, gender-balanced sample of adults, aged ≥18 years at primary and tertiary healthcare facilities in Eastern Cape Province. Of the 1885 participants, the prevalence of CMD, AUD, SMD, SR, and SUD was 24.4%, 9.5%, 8.1%, 6.0%, and 1.6%, respectively. The mwTool-13 yielded high sensitivities for CMD, SMD, and SR, but sub-optimal sensitivities for AUD and SUD (56.7% and 64.5%, respectively). Including a single AUD question in the initial question set improved the tool's performance in identifying AUD and SUD (sensitivity > 70%), while maintaining brevity, face-validity, and simplicity in the South African setting.

## Impact statement

Valid, translated mental health screening tools for non-specialists are necessary for identifying patients with psychiatric disorders in need of critical mental health care. The Mental Wellness Tool-13 (mwTool-13) is a 13-item screener for identifying adults at risk for common and severe mental disorders, alcohol-use and substance-use disorders, and suicide risk. This study validated and improved the mwTool-13 against diagnostic gold standard. The modified SA-mwTool-12 yielded high sensitivities, maintaining brevity, face-validity, and simplicity in the South African setting. Findings from this study support the continued expansion of mental health screening in South Africa at the primary- and community-care level and may inform other validation efforts.

## Introduction

Mental disorders cause substantial disease burden worldwide (Whiteford et al., 2016; GBD 2019 and Mental Disorders Collaborators, 2022; WHO, 2020). This disease burden is disproportionately borne on low- and middle-income countries (LMICs) which lack the psychiatric infrastructure, workforce, and policy to support the high demand for mental health treatment (Alloh et al., 2018; WHO, 2020). Whereby mental disorders increase the risk for other health conditions, such as HIV

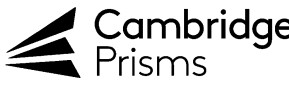

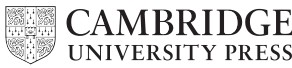

and tuberculosis (TB), and vice versa, there exists the need for integration of services into non-psychiatric settings to combat exacerbated poor health outcomes with comorbid conditions (Prince et al., 2007; Collins et al., 2013; Oh et al., 2017; Sweetland et al., 2018; Hayward et al., 2022). In LMICs, task-sharing mental health care to primary care providers can improve the accessibility of psychiatric services (WONCA, 2008; Javadi et al., 2017; Lovero et al., 2019). Such task-sharing efforts necessitate short mental health screening tools for non-specialists that can identify a broad-spectrum of mental and substance-use disorders (SUDs) and facilitate effective linkage to critical mental health care (Vythilingum et al., 2013; Ali et al., 2016).

In South Africa, lifetime prevalence of any mental disorder in South Africa in 2002 was estimated to be 30.3%; categorized by type of disorder, the lifetime prevalence of anxiety disorders is 15.8%, mood disorders is 9.8%, and alcohol-use disorders (AUDs) or SUDs is 13.4% (Stein et al., 2008). South Africa bears a heavy and unsustainable burden of both TB and HIV individually, and nearly 60% of individuals with TB are also living with HIV (South African National Department of Health, 2021). It is further estimated that one-in-five people living with HIV (PLWH) have a comorbid mental disorder (Myer et al., 2008; Zuma et al., 2022) and approximately one-in-three individuals with TB – with or without HIV coinfection - experience severe psychological distress (Peltzer et al., 2012, 2013; Walt and Moyo, 2018; Janse Van Rensburg et al., 2020). The high rates of mental disorders and infectious diseases, particularly HIV and TB, highlight the need for task-sharing mental health services and validated screening tools.

Unmet need for mental health treatment across the spectrum of mental disorders is high; only one-quarter of South Africans with a mental disorder receive treatment within a given year (Seedat et al., 2008). Lack of specialized providers, inequity in the allocation of both tangible and human resources between provinces, underdeveloped community-based services, and low mental health literacy contribute to the sub-optimal delivery of psychiatric care (Lund et al., 2010; Petersen and Lund, 2011). South Africa has identified mental health task-sharing as a promising strategy to increase access to mental health services while reducing stigma and mental health disparities (Mendenhall et al., 2014).

Brief comprehensive screening for mental disorders is critical to any task-sharing strategy because it enables less-trained providers to identify the presence and severity of mental disorders and make referrals for further clinical evaluation and/or mental health services (Murray et al., 2014). Unfortunately, most screening tools are specific to single disorders, thus requiring multiple screening tools to assess for more than one condition. Such an approach is not optimal, nor is it feasible in under-resourced settings. Moreover, clinical presentations of mental disorders vary in sub-Saharan Africa in comparison to Western settings due to differences in idiomatic descriptions of distress and emotions, and the somatization of psychiatric symptoms (Sweetland et al., 2014; Ali et al., 2016). In South Africa, as in other non-Western countries, there is need for existing screening tools to be linguistically and culturally validated to ensure appropriateness in their patient populations. Additionally, validation in the South Africa epidemiologic setting will allow for reliable identification of psychiatric conditions within a high-burden infectious disease context.

To facilitate broad-spectrum mental and substance-use screening at the primary- and community-care levels, the Mental Wellness Tool-12 (mwTool-12) was recently developed in Mozambique to identify symptoms of common mental disorders (CMDs – depression, anxiety, and post-traumatic stress disorder), severe mental disorders (SMDs – psychosis and mania), AUD, and suicide risk (SR) (Lovero et al., 2021). The mwTool-12 originally included screening questions for SUD, however, the low prevalence of SUD in the Mozambican sample prevented their validation (Lovero et al., 2021). While the mwTool-12 offers substantial utility as a mental and SUD screener in other low-resource settings, the mwTool-12 has yet to be assessed outside of Mozambique. Embedded within a larger study validating a battery of mental health screeners, we sought to evaluate the performance of the mwTool-12 augmented with an additional SUD item – henceforth the mwTool-13 – in Eastern Cape Province, South Africa, against the Mini International Neuropsychiatric Interview (MINI) diagnostic gold standard. For those questions that yielded sub-optimal sensitivity, we sought to improve the mwTool-13's performance while prioritizing high sensitivity, brevity, and face-validity.

## Methods

### Study setting

Data were collected from four primary care clinics within the Buffalo City Metro (BCM) Health District Department of Health in Eastern Cape Province, South Africa from February to April 2022. At these facilities, nurses provide primary care, emergency, and outpatient mental health services, for disorders such as depression, anxiety and posttraumatic stress disorder (PTSD). Specialist Care for more severe mental health conditions is typically rendered in a hospital setting following a referral from a primary or community health facility. In order to capture sufficient numbers of individuals with SMD, additional data were collected from one tertiary care facility in BCM District in May 2022. Eastern Cape Province has a particularly high HIV prevalence (25.2%; 95% CI: 19.8%-31.5%), high TB incidence (1236 per 100,000 persons; 95% CI: 945-1526), and poor HIV, TB, mental health, maternal-child health, and health service delivery indicators (Massyn et al., 2017; National Department of Health, 2018; Massyn et al., 2019; Microbiologically Confirmed Pulmonary TB – Centre for Tuberculosis, 2019; Simbayi et al., 2019; Massyn et al., 2020)

### Study population

Adults (patients and their accompaniers age ≥18 years) at study health facilities were eligible to participate. Individuals were excluded if they were unable to sufficiently communicate in isiXhosa or English.

### Measures

All instruments (the mwTool-13 and MINI) were translated into isiXhosa through a robust process of forward and backward translation, and thorough review by the study investigators, research staff, and local psychiatrist to ensure the face-validity of the instruments.

#### Mental disorder diagnosis and classification

Current mental disorders were diagnosed with the MINI, a structured diagnostic interview that has been widely used as a reference standard across many contexts (Sheehan et al., 1998). With the exception of PTSD and SR, which were diagnosed with the MINI-Plus modules for simplicity and brevity, all other disorders were diagnosed with the relevant MINI-V modules. Details on the small modifications made to the MINI modules can be found in

**Table 1.** MwTool-13 questions definitions of a positive screen for each disorder category

| Numbering, questions and administration instructions | | Definition of a positive screen | Disorder |
|---|---|---|---|
| Step One Questions | 1. In the last 2 weeks, how often have you been feeling down, depressed, or hopeless? | ≥ "Several days" to any of the three questions | CMD |
| | 2. In the last 2 weeks, how often have you been feeling nervous, anxious, or on edge? | | Does not inform a specific disorder[a] |
| | 3. In the last 2 weeks, how often have you been so restless that it's hard to sit still? | | CMD |
| POSITIVE to questions 1 or 2 or 3, CONTINUE SCREENING. If NEGATIVE for all three, STOP. If POSITIVE to question 2, but NEGATIVE to questions 5-11, refer to self-help. | | | |
| Step Two Questions | 4. In the past year, how often do you have a drink containing alcohol? | Anyone (regardless of gender): ≥ "Between 2 and 4 times a month" on Q4 Women: "Monthly or less" on Q4 and ≥ "3 or 4" on Q5 Men: "Monthly or less" on Q4 and ≥ "5 or 6" on Q5 | AUD |
| | 5. In the past year, how many drinks containing alcohol do you have on a typical day when you are drinking? | | |
| | 6. In the past year, how many times have you used a recreational or illegal drug or used a prescription medication for non-medical reasons? | ≥ "Once or twice" | SUD |
| | 5. In the past year, have you ever felt that your thoughts were being directly interfered with or controlled by some outside force or person in a way that many people would find hard to believe (for instance, through telepathy)? | "Yes" to any of the four questions | SMD |
| | 6. In the past year, have there been times when you felt that a group of people was plotting to cause you serious harm or injury? | | |
| | 7. In the past year, have there been times when you felt that something so strange was going on that other people would find it very hard to believe? | | |
| | 8. In the past year, did you at any time hear voices saying quite a few words or sentences when there was no one around that might account for it? | | |
| | 9. In the past month, have you wished you were dead or wished you could go to sleep and not wake up? | "Yes" to any of the three questions | SR |
| | 10. In the past month, have you had any actual thoughts of killing yourself? | | |
| | 11. In the past 3 months, have you ever done anything, started to do anything, or prepared to do anything to end your life? | | |

Abbreviations: AUD, alcohol-use disorder; CMD, common mental disorder; SMD, severe mental disorder; SR, suicide risk; SUD, substance use disorder.
[a]While Q1-3 direct continuation to the step two questions for identifying AUD, SUD, and SMD, positive responses to Q1 and/or Q3 are considered indicative of CMD. Of note, endorsing only Q2 and none of the other questions is not indicative of a specific disorder.

Supplementary Annex 1. We classified participants into the following five categories based on responses to relevant MINI modules:

- SMD: manic episode (mania), hypomanic episode (hypomania), psychotic disorder (psychosis)
- CMD: major depressive episode (depression), PTSD, general anxiety disorder (anxiety)
- AUD: alcohol abuse or dependence
- SUD: substance abuse or dependence
- SR: moderate-to-high SR

### Mental wellness tool-13

The mwTool-13 includes the original mwTool-12 items augmented with one additional SUD item (for a total of 13 items; Lovero et al., 2021; Smith et al., 2010). The mwTool-13 is meant to be administered in two steps. Step 1: patients respond to an initial three questions (Q1-3) to identify those who have any disorder. Step 2: those who endorse any of the initial three items in step one then respond to the remaining 10 questions and are classified into CMD (positive response to Q1 and/or Q3), AUD (positive response to Q4 and/or Q5), SUD (positive response to Q6), SMD (positive response to any of Q7-10), and SR (positive response to any of Q11-13) groups. The mwTool-12 was developed using a data-driven item-selection method; these 12 items were identified from

a battery of 99 items across nine commonly used mental disorder and functioning assessments (Bebbington and Nayani, 1995; American Psychiatric Association, 2000; Spitzer et al., 2006; Posner et al., 2011; Prins et al., 2016) using a variable selection technique, the least absolute shrinkage and selection operator (LASSO) (Tibshirani, 1996). The data-driven selected items were then reviewed by clinical experts, who confirmed the first three items both captured symptoms of CMDs as well as reflected comorbid symptoms with the other disorders. We augmented the original mwTool-12 by adding an additional SUD item (Smith et al., 2010). See Supplementary Annex 2 for MwTool-13 questions and response options, and Table 1 for definitions of a positive screen for each disorder. To ensure proper evaluation of the mwTool-13 performance, participants responded to all 13 questions regardless of their responses to the step one question set.

### Demographic and general health measures

We collected self-reported sociodemographic information (age, gender, marital status, living situation, education, religion, monthly household income, occupation, and ethnicity), physical health history (non-communicable diseases, pregnancy, and parity), and mental health history (prior mental health diagnosis, prior access to mental healthcare).

### Data collection & procedures

Over a 2-week period, a team of five research assistants (RAs) were trained to administer the mwTool-13, and eight nurses were trained to administer and confirm psychiatric diagnoses using the MINI. The RAs were all experienced community health workers. The nurses had all received psychiatric training during their nursing formation and had over 10 years of clinical experience. Study staff piloted data collection tools over a 2-week prior to study start. During the pilot period, study staff met with a local psychiatrist to debrief and discuss any concerns or issues with the diagnostic interview. During the review process, challenges were identified in diagnosing SMD, particularly given the overlap between symptoms of psychosis and accepted cultural norms. The nurses were empowered to use their clinical acumen to differentiate between abnormal and culturally normative beliefs.

To reduce recruitment bias and ensure a representative sample of adults at the health care facilities, RAs were instructed to approach every seventh person entering the collaborating health facility. Potential participants were screened for their age and informed about the study. Interested individuals were read the informed consent form in an erected gazebo which provided privacy. The RAs then administered the sociodemographic questionnaire and mwTool-13 to consenting participants. The nurses – blinded to the results of the screening – then administered the MINI in a separate gazebo. Bilingual staff conducted all study activities in a participants preferred language (i.e., English or IsiXhosa). Responses to all measures were recorded using REDCap on a tablet computer (Harris et al., 2009).

Individuals with MINI-diagnosed, psychiatric disorders were managed according to facility policies and South African national guidelines (Petersen et al., 2016). This included referral to existing psychiatric staff; in the absence of specialized psychiatric services in line with the facility Operational Managers purview, the nurses used the South African Adult Primary Care (APC), which is mean to guide clinical decisions, and the Integrated Chronic Disease Management (ICDM) manuals to link individuals to the necessary services (Department of Health, 2014, 2019).

Our target sample included 50 gender-balanced individuals per disorder (depression, anxiety, PTSD, AUD, SUD, Psychosis, Mania (hyper or hypo), SR) and at least 100 individuals without any disorder in order to obtain precise confidence intervals. Given the higher rate in which women attended our study clinics, RAs were instructed to target all available men to ensure gender-balanced sampling,

### Statistical analysis

Participants with incomplete responses to the MINI or mwTool-13 were excluded from analysis. We used descriptive statistics to describe the study population and present the prevalence of confirmed MINI-diagnoses. Sensitivities, specificities and 95% confidence intervals (CIs) were calculated for each of the disorder categories using the original two-step administration and the original definition of a positive-case (see Table 1 for Original Definitions). For disorder categories in which the two-step approach and original definition did not yield adequately high sensitivities (>70%), we explored changing the definition of a positive screen and including the questions in the step one questions set. We performed additional analyses restricted to those who received the mwTool-13 in isiXhosa and then stratified analyses with respect to gender, HIV status, and lifetime TB history.

### Ethical considerations

The study was approved by the New York State Psychiatric Institute Institutional Review Board (Protocol #8272), the Foundation for Professional Development Research Ethics Committee (8/2021) and the Eastern Cape Department of Health Research Committee (EC_202110_015).

## Results

### Participant characteristics

The 1885 participants' socio-demographic characteristics are presented in Table 2. The average age of participants was 39 years, 65% of participants were female, nearly all participants identified as black (97.3%), and 97.4% of participants reported isiXhosa as their dominant language. Nearly all (95.1%) participants received the mwTool-13 in isiXhosa. Of note, 72.4% were seeking health services for themselves, while 27.6% were accompanying someone seeking care.

At enrollment, participants were asked to self-report current and prior diagnoses of communicable and non-communicable diseases, including TB, HIV and mental disorders (Table 2). Notably, 30.6% reported having a diagnosed non-communicable disease (hypertension, diabetes, epilepsy, asthma or other), 15.1% reported a current or previous bout of TB and 25.8% reported they were living with HIV. Of the 268 participants who reported a previously diagnosed mental health disorder (depression, anxiety, PTSD, bipolar disorder, panic, suicidality, alcohol or substance abuse, or schizophrenia), 85% (*n* = 228) reported receiving treatment for a mental disorder.

### Prevalence of confirmed, MINI-diagnosed mental disorders

The prevalence of confirmed, MINI-diagnosed mental and SUDs is presented in Table 3. Specifically, 36% of participants were diagnosed with at least one disorder: CMD, 24.4%; AUD, 9.5%; SMD, 8.1%; SR (moderate to high), 6%; and SUD, 1.6%. While we achieved our gender-balanced target for CMD, AUD, SMD and SR, we did not for SUD. Of the 673 with at least one diagnosis; 53.8% (*n* = 362) had more than one diagnosis, of the 458 with a CMD diagnosis, 38.0% (*n* = 174) had more than one cmd. Of the 205 of with either AUD or SUD, *n* = 6 had both.

### Performance of the MwTool-13 (initial 3 questions + 10 questions)

Using the original two-step administration method, the mwTool-13 performed well as evidenced by strong sensitivity for identifying any disorder (83.33%), CMD (91.50%), SMD (71.71%), and SR (86.84%) (Table 4). However, the AUD questions (Q4 and Q5) and the SUD question (Q6) yielded sub-optimal sensitivity using the two-step method (56.67% and 64.52%, respectively). Results restricted to those who received the mwTool-13 in isiXhosa and results stratified by gender, HIV status, and lifetime TB history are available in Supplementary Annex 3. The tool similarly for all subgroups. There are some differences in performance for identifying any disorder, CMD, depression and AUD by gender, with generally higher sensitivity and lower specificity for women when compared to men.

**Table 2.** Participant characteristics (*N* = 1885), by gender

| Mean (SD) or n (%) | Total (*N* = 1885) | Men (*n* = 651) | Women (*n* = 1232) |
|---|---|---|---|
| Age (range: 18–88) | 39 (13.1) | 39.4 (12.4) | 38.9 (13.5) |
| Perinatal | | | |
| Currently pregnant | – | – | 90 (7.3) |
| Gave birth in past 12–months | – | – | 127 (10.3) |
| Marital status | | | |
| Single | 1114 (59.1) | 409 (62.8) | 703 (57.1) |
| In a relationship/married | 624 (33.1) | 111 (17.1) | 209 (17) |
| Separated, divorce, widowed | 147 (7.8) | 43 (6.6) | 104 (8.4) |
| Dominant language | | | |
| isiXhosa | 1836 (97.4) | 632 (97.1) | 1202 (97.6) |
| Afrikaans | 30 (1.6) | 11 (1.7) | 19 (1.5) |
| English | 15 (0.8) | 7 (1.1) | 8 (0.6) |
| Other | 4 (0.2) | 1 (0.2) | 3 (0.2) |
| Language of mwTool-13 Administration | | | |
| English | 92 (4.9) | 26 (4) | 66 (5.4) |
| isiXhosa | 1792 (95.1) | 625 (96) | 1165 (94.6) |
| Interchangeable | 1 (0.1) | 0 (0) | 1 (0.1) |
| Race | | | |
| White | 17 (0.9) | 7 (1.1) | 10 (0.8) |
| Black | 1834 (97.3) | 634 (97.4) | 1198 (97.2) |
| Indian | 1 (0.1) | 0 (0) | 1 (0.1) |
| Colored | 31 (1.6) | 10 (1.5) | 21 (1.7) |
| Other | 2 (0.1) | 0 (0) | 2 (0.2) |
| Education | | | |
| Grade 7 (primary) or less | 219 (11.6) | 73 (11.2) | 146 (11.9) |
| Grade 8-11 (before matric) | 733 (38.9) | 264 (40.6) | 468 (38) |
| Grade 12 (matric) | 745 (39.5) | 256 (39.3) | 489 (39.7) |
| Tertiary | 188 (10) | 58 (8.9) | 129 (10.5) |
| Residence | | | |
| Informal dwelling | 551 (29.2) | 171 (26.3) | 380 (30.8) |
| Formal house | 1329 (70.5) | 478 (73.4) | 849 (68.9) |
| Other | 5 (0.3) | 2 (0.3) | 3 (0.2) |
| Income | | | |
| < R1000 (~57 USD) | 1061 (56.3) | 347 (53.3) | 714 (58) |
| R1000-R5000 (~57-284 USD) | 656 (34.8) | 227 (34.9) | 429 (34.8) |
| >5000 (~284 USD) | 168 (8.9) | 77 (11.8) | 89 (7.2) |
| Employment | | | |
| Unemployed | 1061 (56.3) | 347 (53.3) | 714 (58) |
| Employed | 656 (34.8) | 227 (34.9) | 429 (34.8) |
| Student | 168 (8.9) | 77 (11.8) | 89 (7.2) |

(Continued)

**Table 2.** (*Continued*)

| Mean (SD) or n (%) | Total (*N* = 1885) | Men (*n* = 651) | Women (*n* = 1232) |
|---|---|---|---|
| Unpaid volunteer, self-employed, Other | 1061 (56.3) | 347 (53.3) | 714 (58) |
| Health seeking | | | |
| Seeking care | 1364 (72.4) | 460 (70.7) | 902 (73.2) |
| Accompanying | 521 (27.6) | 191 (29.3) | 330 (26.8) |
| Self-reported Health Status | | | |
| Non-communicable disease (Any) | 577 (30.6) | 176 (27) | 401 (32.5) |
| Mental disorder (Any) | 268 (14.2) | 135 (20.7) | 133 (10.8) |
| Ever had TB | 284 (15.1) | 131 (20.1) | 153 (12.4) |
| Positive HIV status | 487 (25.8) | 119 (18.3) | 366 (29.7) |

Note: Two participants reported they were trans/non-binary for "gender."

## Performance of the modified mwTool – SA-mwTool-12

Due to the sub-optimal performance of substance-use questions using the original administration and definitions of a positive screen, we proposed the following modifications. We included the AUD questions in the step one question set, which improved sensitivity. We dropped the drinking frequency question (Q4) as it lengthened the questionnaire without improving the sensitivity. We changed the definition of positive screen for AUD as ≥ 3 or 4 drinks on the drinking amount question (Q5) regardless of gender, which upheld face-validity. A more detailed description of steps considered in arriving at the proposed modification is available in Supplementary Annex 4.

The modified mwTool-13 – henceforth the SA-mwTool-12 – thus included a modified initial set of four questions [Q1–Q3 + Q5 (Drinking Amount, ≥ 3 or 4)], where only those who endorse Q1–Q3 and/or Q5, would receive the remaining eight questions (Q6–Q13). The SA-mwTool-12 yielded strong sensitivity for identifying any disorder (89.66%), CMD (91.48%), AUD (74.44%), SUD (74.19%), SMD (73.02%), and SR (88.60%) (Table 5). Results restricted to those who received the mwTool-13 in isiXhosa and results stratified by gender, HIV status, and lifetime TB history showed similar performance and are available in Supplementary Annex 5. The final SA-mwTool-12, definitions of a positive screen and administration instructions can be found in Supplementary Annex 6.

## Discussion

In this validation study, the mwTool-13 performed as well as the mwTool-12 screener for identifying any disorder, CMD, SMD, and SR (Lovero et al., 2021), but sub-optimally for AUD and SUD. However, the performance of the mwTool-13 was significantly improved by adding a single AUD question (about drinking amount) to the step one question set and defining of a positive screen for AUD as ≥3–4 drinks. By training RAs to administer the mwTool-13, we have demonstrated an ability to build the mental health screening competency of lay-health workers. This supports evidence that community-level health workers can be trained to provide evidence-based mental health screening in low-resource

**Table 3.** Prevalence of MINI-diagnosed mental and substance use disorders diagnosed, by gender

| n(%) | Total (N=1885) | Men (n=651) | Women (n=1232) |
|---|---|---|---|
| Any disorder | 673 (35.7) | 239 (36.7) | 433 (35.2) |
| CMD | 458 (24.3) | 107 (16.4) | 350 (28.4) |
| Major depressive episode | 406 (21.5) | 93 (14.3) | 312 (25.3) |
| Generalized anxiety disorder | 121 (6.4) | 23 (3.5) | 97 (7.9) |
| PTSD | 163 (8.6) | 30 (4.6) | 132 (10.7) |
| AUD | 180 (9.5) | 75 (11.5) | 105 (8.5) |
| SUD | 31 (1.6) | 26 (4) | 5 (0.4) |
| SMD | 152 (8.1) | 78 (12.0) | 74 (6.0) |
| Psychotic disorder | 95 (5) | 66 (10.1) | 29 (2.4) |
| Hypomanic episode | 47 (2.5) | 13 (2.0) | 34 (2.7) |
| Manic episode | 20 (1.1) | 7 (1.1) | 13 (1.1) |
| SR | 114 (6) | 20 (3.1) | 94 (7.6) |
| Moderate risk | 33 (1.8) | 7 (1.1) | 26 (2.1) |
| High risk | 81 (4.3) | 13 (2) | 68 (5.5) |

Abbreviations: AUD, alcohol-use disorder; CMD, common mental disorder; SMD, severe mental disorder; SR, suicide risk; SUD, substance-use disorder.

settings (Wainberg et al., 2017). Furthermore, by validating the SA-mwTool-12, this study adds to the nascent, but growing collection of translated, valid mental health screeners for the South African setting.

As evidenced by findings from this study, there is a large burden of mental health disorders in South Africa and an unmet need for mental health services (Seedat et al., 2008; Herman et al., 2009). Given the limited resources to treat individuals with mental disorders, valid screening tools are needed in low-resource settings such as South Africa (Kagee et al., 2013). One means of achieving valid screeners is to ensure that tools are available in the local languages spoken by people accessing public healthcare services; South Africa has 11 official languages, with isiXhosa spoken primary language of 16% of South Africans and predominately spoken in Eastern Cape Province (Statistics South Africa, 2012). Very few studies have translated and validated mental health screeners into isiXhosa or in South Africa at large. The translated isiXhosa mwTool-13 assists in filling this gap and helps provide the necessary tools for the continued expansion of mental health services in South Africa.

While the SA-mwTool-12 yielded high sensitivities for identifying CMD, the specificities ranged from 41.5% to 47.7%. The SA-mwTool-12 is not designed to function as a diagnostic tool, but to serve as a primary- and community-care level tool to facilitate early detection and intervention. Thus, higher sensitivity is prioritized to ensure those who require further screening and potential intervention are identified. Follow-up assessment to confirm mental health distress or diagnosis would be warranted. Future programing and implementation research using the SA-mwTool-12 should recognize the potential for over-identification of individuals in need of additional assessment and the impact this could have on already strained mental healthcare system.

Hazardous alcohol and drug use are stigmatized behaviors (Sorsdahl and Stein, 2010; Sorsdahl et al., 2012; Van Boekel et al.,

2013; Zewdu et al., 2019; Regenauer et al., 2020; Magidson et al., 2022), potentially due to low mental health literacy and criminalization. As such, it is possible that stigma reduced disclosure of alcohol and substance use, as has been documented elsewhere in the region (Hahn et al., 2016). In our study, some participants reported hazardous alcohol and drug use on the mwTool-13 screening, but not when asked about these behaviors during the MINI diagnostic interview (and vice-versa). There is a clear need to understand any reluctance to accurately report substance-use behaviors, as well as why the AUD and SUD screening and MINI questions are being answered or interpreted differently. Cognitive interviewing to systematically understand how participants interpret specific screening items has been identified as a crucial method for reducing measurement error in SUD assessment items (Boness and Sher, 2020). Future research should include cognitive interviewing with the AUD and SUD questions from the mwTool-12 as well as the MINI, particularly when responses are incongruent.

We recommend further validating the SUD question of the mwTool-13 as well as other SUD brief screeners among a population with a higher prevalence of SUD in SA to better understand how that question performs, and explore other appropriate options. The low prevalence (1.6%) of SUD hampered our ability to draw conclusions about the performance of the SUD question (Q6). While limited recent SUD prevalence data exists for South Africa, a behavioral study found the past 3-month prevalence of any drug use was 4.4% and 4.1% in South Africa and the Eastern Cape, respectively (Peltzer and Phaswana-Mafuya, 2018). Another small study conducted among patients receiving care at the psychiatric unit of a hospital in East London, South Africa found that 17.4% of participants reported past year use of psychoactive substances (Tindimwebwa et al., 2021). Recognizing that reporting any use is not equivalent to a SUD diagnosis, it is likely that substance use was under-reported in our sample. Further, research has shown hazardous alcohol use is often comorbid with drug use (Pengpid et al., 2021). Yet, only 0.4% of our total study population (and only 3% of the 180 participants with AUD) were diagnosed with both SUD and AUD. Thus, those with SUD would likely be missed if only AUD screening were offered, demonstrating a need to screen not only for AUD but specifically for SUD as well.

Beyond validating a mental health screening tool, this study yielded many other practical insights for implementing mental health programming in resource-limited settings. First, the modified mwTool-13 offers a practical means to "bundle" screening (a process for simultaneously assessing multiple behavioral health disorders; Mulvaney-Day et al., 2018). In resource-limited settings, most existing validated screening tools identify only a single disorder. As administering a multitude of screeners to identify broad-spectrum mental and SUDs would be impractical and time consuming, the mwTool-13 offers an efficient and effective alternative. The recommended modifications also serve to both shorten the tool and simplify administration. Further, changing the definition of a positive screen for AUD such that it no longer requires a gender-binary classification may facilitate more inclusive, gender-affirming care (Arellano-Anderson and Keuroghlian, 2020). This study also yielded real-world multi-lingual validation data. By allowing participants to choose and vacillate between isiXhosa and English, we were able to validate the mwTool-13 in multiple languages simultaneously. In places that have great ethnic and linguistic diversity, such as South Africa, multi-lingual adaptation and validation of mental health screening is lagging (Kaiser et al., 2019; Kaiser et al., 2022). Ultimately, this study provides both the

**Table 4.** Performance of the mwTool-13

| Disorder | Questions | Mini prevalence | Sensitivity (95% CIs) | Specificity (95% CIs) |
|---|---|---|---|---|
| *Step One Questions* | | | | |
| Any disorder | Q1-Q3 | 36% | 83.36 (80.32-86.10) | 46.45 (43.61-49.31) |
| CMD | Q1-Q3 | 24% | 91.48 (88.54-93.87) | 44.57 (41.97-47.19) |
| CMD | Q1, Q3 | 24% | 88.86 (85.62-91.60) | 47.72 (45.10-50.35) |
| Major depressive episode | Q1, Q3 | 21% | 90.89 (87.66-93.50) | 46.99 (44.42-49.57) |
| Generalized anxiety disorder | Q1, Q3 | 6% | 90.08 (83.32-94.77) | 40.82 (38.51-43.15) |
| PTSD | Q1, Q3 | 9% | 89.57 (83.83-93.81) | 41.52 (39.18-43.89) |
| *Step Two Questions* | | | | |
| AUD | Q4, Q5[a] | 10% | 56.67 (49.09-64.02) | 84.5 (82.69-86.19) |
| SUD | Q6 | 2% | 64.52 (45.37-80.77) | 93.42 (92.19-94.51) |
| SMD | Q7-Q10 | 8% | 71.71 (63.84-78.71) | 61.92 (59.58-64.21) |
| Psychotic disorder | Q7-Q10 | 5% | 67.37 (56.98-76.64) | 60.61 (58.31-62.89) |
| Manic or hypomanic episode | Q7-Q10 | 4% | 77.61 (65.78-86.89) | 60.56 (58.27-62.82) |
| SR | Q11-Q13 | 6% | 86.84 (79.23-92.44) | 80.24 (78.3-82.07) |

Abbreviations: AUD, alcohol-use disorder; CMD, common mental disorder; SMD, severe mental disorder; SR, suicide risk; SUD, substance-use disorder.
[a]Those who did not self-identify as male or female are treated as missing as these definitions are gender dependent (*n* = 2).

**Table 5.** Performance of the SA-mwTool-12

| Disorder | Questions | Mini prevalence | Sensitivity (95% CIs) | Specificity (95% CIs) |
|---|---|---|---|---|
| *Step One Questions* | | | | |
| Any disorder | Q1-Q3, Q5 | 36% | 89.75 (87.20-91.93) | 38.03 (35.29-40.83) |
| CMD | Q1-Q3 | 24% | 91.48 (88.54-93.87) | 44.57 (41.97-47.19) |
| CMD | Q1, Q3 | 24% | 88.86 (85.62-91.60) | 47.72 (45.10-50.35) |
| Major depressive episode | Q1, Q3 | 21% | 90.89 (87.66-93.50) | 46.99 (44.42-49.57) |
| Generalized anxiety disorder | Q1, Q3 | 6% | 90.08 (83.32-94.77) | 40.82 (38.51-43.15) |
| PTSD | Q1, Q3 | 9% | 89.57 (83.83-93.81) | 41.52 (39.18-43.89) |
| AUD | Q5 | 10% | 74.44 (67.42-80.64) | 75.54 (73.43-77.57) |
| *Step Two Questions* | | | | |
| SUD | Q6 | 2% | 74.19 (55.39-88.14) | 91.69 (90.34-92.91) |
| SMD | Q7-Q10 | 8% | 73.02 (65.24-78.89) | 60.99 (58.65-63.29) |
| Psychotic disorder | Q7-Q10 | 5% | 68.42 (58.08-77.58) | 59.66 (57.35-61.95) |
| Manic or hypomanic episode | Q7-Q10 | 4% | 79.10 (67.43-88.08) | 59.63 (57.32-61.89) |
| SR | Q11-Q13 | 6% | 88.60 (81.29-93.79) | 79.39 (77.43-81.25) |

Abbreviations: AUD, alcohol-use disorder; CMD, common mental disorder; SMD, severe mental disorder; SR, suicide risk; SUD, substance-use disorder.

necessary validation data as well as formative practical evidence for future implementation science research into how best to integrate routine broad-spectrum screening into primary and community care in South Africa.

## Limitations

There are some inherent limitations to this study. The original mwTool-12 did not include a question specifically for SUD, and a more robust effort to augment the tool prior to launching the validation exercise may have been warranted. Even though the mwTool-12 is meant to be used in primary and community care settings, the study sample consisted of a targeted sample of adults present at primary and tertiary care facilities. As such, the prevalence data reported in this manuscript should not be interpreted as generalizable or representative of the South African adult population and we did not calculate the positive predictive value or negative predictive value of the tool. We did not have a large enough sample of individuals who were screened in English to separately validate the screener in English. We also administered the mwTool-13 screening prior to the MINI, which may have biased the MINI diagnostic interviews.

## Conclusions

The mwTool-13 yielded high sensitivities for identifying CMD and SR. The recommended modifications to the mwTool-13 improved the tool's performance in identifying AUD, SUD, and SMD while maintaining brevity in the South African setting. However, further research into appropriate screening for harmful substance use is warranted. The resulting SA-mwTool-12 offers a valid, translated and culturally relevant brief screening measure for broad-spectrum disorders in South Africa and other low-resource settings. Findings from this study support the continued expansion of mental health screening in South Africa at the primary- and community-care level, facilitate access to appropriate mental health services, and may inform other validation efforts.

**Open peer review.** To view the open peer review materials for this article, please visit http://doi.org/10.1017/gmh.2023.89.

**Supplementary material.** The supplementary material for this article can be found at http://doi.org/10.1017/gmh.2023.89.

**Data availability statement.** The de-identified data may be made available upon reasonable request.

**Acknowledgements.** We express our gratitude to the data collectors and study participants without whom this study would not be possible.

**Author contribution.** M.A.S. designed the study, analyzed the results, and drafted the manuscript. E.W.M. and L.M. lead the study staff training and help develop data collection protocols, with support from K.N., N.N., A.C.W., P.N., C.B., and C.G. C.G. lead the clinical and provide clinical oversight throughout the course of the study. C.B. and M.M.W. provide statistical expertise and consultation. C.B. developed the data collection system. G.S., K.L.L., D.O. drafted portions of the manuscript. M.L.W., A.M., and P.N. provided senior leadership and oversight.
All authors contributed to the drafting of this manuscript.

**Financial support.** This study was supported by National Institute of Mental Health grant U19MH113203, MAS was additionally supported by T32MH096724 and K01MH130226, and KLL was supported by K01MH120258. EWM was supported by the University of California Fogarty GloCal Health Fellowship.

**Competing interest.** The authors declare there are no conflicts of interest.

**Ethics standard.** The study was approved by the New York State Psychiatric Institute Institutional Review Board (Protocol #8272), the Foundation for Professional Development Research Ethics Committee (8/2021) and the Eastern Cape Department of Health Research Committee (EC_202110_015).

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
