## [Reviewer Report]

May 9th, 2023

Submission of manuscript for review, Global Mental Health

Dear Drs. Bass and Chibanda,

I would be grateful if you would consider the following research article by Stockton et al. “Validation of a Brief Screener for Broad-Spectrum Mental and Substance Use Disorders in South Africa” for publication in Global Mental Health. In this manuscript, we present findings from a validation study evaluating the performance of the Mental Wellness Tool-13 in Eastern Cape, South Africa against a diagnostic gold standard. We believe our manuscript will support the continued expansion of mental health screening in South Africa at the primary- and community-care level, facilitate access to appropriate mental health services, and may inform other validation efforts in low-resource settings. If published, it would add to the nascent, but growing, arsenal of translated and validated mental health screeners for the South African setting. 

Finally, we believe that our manuscript complements other papers published in the journal in recent years on mental health screeners and validated tools: 

• Abrahams and Lund (2022) “Food insecurity and common mental disorders in perinatal women living in low socio-economic settings in Cape Town, South Africa during the COVID-19 pandemic: a cohort study”

• Bitta et al. (2022) “Validating measures of stigma against those with mental illness among a community sample in Kilifi, Kenya” 

• Spedding et al. (2022) “ENhancing Assessment of Common Therapeutic factors (ENACT) tool: adaption and psychometric properties in South Africa”

We confirm that this manuscript has not been published elsewhere and is not under consideration by another journal. All authors have approved the manuscript. Should you have any query, please do not hesitate to contact me via email. 

On behalf of all authors, I look forward to hearing from you soon.

Yours Sincerely

Melissa Stockton, PhD

Assistant Professor

Department of Epidemiology

UNC-Chapel Hill

---

## [Reviewer Report]

This is an interesting and well-written manuscript. Developing validated screening tools for mental health in LMICs is an important research endeavour as it has the potential to significantly contribute to the development of measures that are appropriate for these contexts.

Overall, the screening for mental wellness mw Tool-13 records good sensitivity (true positives), but has unacceptable specificity outcomes (true negatives). The abstract indicates that “The mwTool-13 performed well for CMD, SMD, and SR, but sub-optimally for AUD and SUD (sensitivities of 56.7% and 64.5%, respectively).” Unfortunately, this statement is only partly true and the abstract should more accurately report the low specificity values for CMDs and variable performance of SMDs and low sensitivities for AUDs and SUDs to further develop the screening tool.

Introduction

An important argument in the paper is the need to develop mental health screening tools for low-resource contexts. The paper specifically emphasizes high sensitivity alongside a broad spectrum of psychiatric conditions. How do the authors envisage that this could happen given the focus on sensitivity which leads to an over-identification of patients, potentially overwhelming meagre mental health services, especially in LMIC contexts?

Page 3 (lines 43-52). The argument would logically read better if the paragraph began with lines 47-52, followed by lines 43-46.

Methods

General comment - it would be helpful to identify Tables that are meant to be read in conjunction with the text given that there are additional annexes and tables!

Page 4: Greater detail is needed about the level of training and characteristics of a health care worker who provides “outpatient mental health care services”?

Page 5 (Line 107): “... thorough review by the study investigators, research staff, and local psychiatrist.” Was the purpose of the review to establish the face validity of the measure? Did the procedure adopt a formal review process?

Page 5 (Line 128): It would be useful to have greater clarity on the conceptual basis of the three “filter” questions given that two questions refer to CMDs while a third is deemed to be non-specific, though it could easily be considered to be a measure of an anxiety disorder! The link between these and any substance use or alcohol use disorder or severe mental health disorder is not immediately apparent.

Page 6: (Line 154): Could the authors explain how trained RA’s are equivalent to community health workers? What is the length of training received by the RA’s and nurses and how was fidelity established? What are the demographic characteristics of the RAs and nurses trained in the MINI? For example, did the nurses have any psychiatric training, how long have they been in practice etc? What were the major concerns or issues that emerged in the pilot phase?

Page 7: (Line 172-173). Could the authors please clarify why the ICSM (which is part of an ideal clinic practice) was used instead of Adult Primary Care (APC) given that the former determines policies and procedures of how referrals are to be made at a facility level, while the APC manual is concerned with everyday clinical management of patients as part of DoH policy? The APC is used to further interrogate screening decisions by a primary care nurse.

Page 8 (line 211) and Table 2 (Line 9). While 14% (n=268) of the respondents had a previously diagnosed mental disorder, an astonishing 85% (n=228) of these received treatment. This is unusual in a scare-resource context where the number of mental health service providers is limited. Are there any details about who provided such services and was this at the time of the visit or at some other time?

Results

It would be useful to know about the reliability of the overall or specific elements of the mwTool. If this statistic is deemed to be unimportant, please provide an explanation.

In establishing the utility of a screening tool, an important statement of the accuracy of the screening tool is its relative performance in relation to positive and negative predictive values in identifying those with and without a mental health disorder. Is there an explanation as to why this was not considered in the analysis?

Table 3 shows the prevalence of MINI-diagnosed individuals. Given that the total number exceeds the total recruited, please clarify how many participants had multiple disorders.

Tables 4 and 5 do not indicate a similar prevalence profile making it difficult to understand the relative performance of the mwTool-13/ 12 to that of the MINI. In essence, did the mwTool-13/ 12 identify a similar number of individuals in each of the diagnostic categories as did the MINI?

Tables 2, 3 and 4 all show that AUD questions had better specificity ratios than sensitivity ratios. Could the authors speculate about possible reasons for this change in direction relative to the rest of the mwTool?

Discussion:

Page 10 (line 257). Overall, given the low specificity of the mwTool-13 on CMDS, low sensitivity on AUD and SUD and variable performance on SMDs, the claim that the mwTool as it stands is a valid mental health screening tool will need more substantial justification. Nevertheless, the authors recognise that further work is necessary for the mwTool to be generalizable to other settings and population groups.

Page 10 (Lines 261-268): The authors argue for high specificity in low mental health resource contexts in LMICs. This is confusing as the study deliberately sets out to demonstrate the validity of the mwTool. The argument for specificity then is conflated with making the tool available in isiXhosa. Given the poor performance of the mwTool in relation to specificity, except for AUD and SUD subscales, the intention behind this argument should be clarified.

Page 10 (Lines 270-272). It is argued that hazardous alcohol and drug use are stigmatized and have the impact of reducing disclosure. It is unclear why substance use disorders should be more stigmatizing than having a common mental disorder or a severe mental disorder. Further, the explanation that the low specificity ratios are a function of low numbers is somewhat negated as they are also low for CMDs and SMDs among those who received the tool in isiXhosa, by gender, self-reported HIV status and self-reported lifetime TB history! Those attending the clinic may be aware that reporting SUDs will likely result in a referral. This might explain why the number of SUDs self-reports was low. If participants’ reasons for visiting the facility are available, these might be useful to include in the table on demographics, supporting the argument that most attend such facilities not seeking assistance with AUDs or SUDs.

The discussion about the non-performance of SUD items on the mwTool is disproportionate to the discussion of the performance of subscales on the mwTool, some of which performed poorly as a measure of sensitivity (AUD; Psychotic Disorder).

Conclusions

Page 12 (line 323). The statement that “the mwTool-13 performed well” needs moderation as the mwTool performed well concerning sensitivity on CMDs but not SUDS and AUD which had higher specificity ratios and variable performance on Psychotic Disorders subscale.

Minor Comments:

Page 2 (line 31) sentence needs fixing

Page 2 (line 33) - Reference is incorrect

---

## [Reviewer Report]

Impact statement

The impact statement s written well.

Abstract

The abstract is well structured.

1introduction

Although a rationale is provided for the study, changes that were made to the Mozambique tool are not justified and the rationale of that is not clear. Given the importance of substances use disorders in South Africa and other sub-Saharan Africa, an attempt should have been made to achieve a broader tool. Additional questions were included, but this was not associated with useful psychometric properties for example, a sensitivity of about 77%.

2 Methods

2.1 Study settings

Screening for CMD, SUD etc would ideally be done in primary care settings and so the tool would have been validated in this population. However, the researchers validated the tool in both primary and tertiary settings. this might have affected the outcomes in terms of the achieved psychometric properties as well as reducing its usefulness in the primary care settings.

2.2 Study population

The study population is described well and is appropriate to achieve the aims of the study.

2.3 Measures

The tools used in the study are described well. the choice of the gold standard is justified.

2.6 Ethical considerations

The ethical considerations are well described.

3 Results

All the results are presented well. The sensitivities achieved for the tools are promising and the tool is good enough for assessing CMD, but not helpful for SUD and AUD. This is of concern given the comorbidities of SUD and AUD.

4 Discussion

The discussion is presented well.

Limitations

The study limitations are well presented.

5 Conclusion

Although the conclusions are presented well, the challenges of assessing AUD and SUD need to be attended to well. A tool that will be useful for CMD, Severe mental illness, SUD and AUD are important. This study did not produce such and this is an important limitation.

---

## [Reviewer Report]

Thank you for submitting your manuscript on “Validation of a Brief Screener for Broad-Spectrum Mental and Substance Use Disorders in South Africa”. We can, unfortunately not accept the manuscript in its current form. Kindly attend to the reviewers' comments that are included in this email.

---

## [Reviewer Report]

Dear Editors, 

We greatly appreciate your thoughtful reviews. We have responded to the comments as requested and made all changes to the manuscript in tracked changes. Of note, we noted a discrepancy in how we had captured current severe mental disorders. As such, we made some changes to the tables and result section. Our detailed responses to your comments can be found in the response to reviewers box. 

Sincerely, 

Authors

---

## [Reviewer Report]

The authors have made significant and appropriate changes to the MS. A concern is the use of the term “performed well” in the MS as a whole. While the measure attained good sensitivities on most of the subscales, it did not do so on all the subscales. The specificities were also poor even though the intention was to obtain high sensitivity indices. On this basis, the use of the phrase “performed well” is misleading. The following issues need to be addressed or modified as appropriate:

Page 1: Line 5: Delete “performed well” in the sentence as it appears to be superfluous

Page 2: Line 32: Impact statement: The term “performed well” should be removed.

Page 3: Line 34: Facilitating access to appropriate mental health services was not tested and therefore should not be part of the impact statement.

Page 7: Line 196-197: It may be useful to indicate that for clinical purposes, it is the APC that has to be used.

Page 9: Line 247: Typo

Page 12: Line 362: Typo

Page 12: Line 363: The term “performed well” does not fit with the description of a measure that was only partially successful regarding sensitivity estimates. It tends to overstate the outcomes of this study which is nevertheless valuable.

---

## [Reviewer Report]

Dear Editors, 

We greatly appreciate your thoughtful reviews. We have responded to the comments as requested and made all changes to the manuscript in tracked changes. Our detailed responses to your comments can be found below each comment.

Sincerely, 

Authors

Reviewer: 1

Comments to the Author

Thank you authors for attending to my concerns.

Reviewer: 2

Comments to the Author

The authors have made significant and appropriate changes to the MS. A concern is the use of the term “performed well” in the MS as a whole. While the measure attained good sensitivities on most of the subscales, it did not do so on all the subscales. The specificities were also poor even though the intention was to obtain high sensitivity indices. On this basis, the use of the phrase “performed well” is misleading. The following issues need to be addressed or modified as appropriate:

We removed “performed well” from the entire manuscript, and have instead highlighted the high sensitivities

Page 1: Line 5: Delete “performed well” in the sentence as it appears to be superfluous

We removed “performed well” from the entire manuscript.

Page 2: Line 32: Impact statement: The term “performed well” should be removed.

We removed “performed well” from the impact statement.

Page 3: Line 34: Facilitating access to appropriate mental health services was not tested and therefore should not be part of the impact statement.

We have removed this from the impact statement

Page 7: Line 196-197: It may be useful to indicate that for clinical purposes, it is the APC that has to be used.

We have indicated the APC is meant to guide clinical decisions.

Page 9: Line 247: Typo

Corrected

Page 12: Line 362: Typo

Corrected

Page 12: Line 363: The term “performed well” does not fit with the description of a measure that was only partially successful regarding sensitivity estimates. It tends to overstate the outcomes of this study which is nevertheless valuable.

We have removed “performed well.”